# High-Efficiency Achromatic Metalens Topologically Optimized in the Visible

**DOI:** 10.3390/nano13050890

**Published:** 2023-02-27

**Authors:** Lijuan Zhang, Chengmiao Wang, Yupei Wei, Yu Lin, Yeming Han, Yongbo Deng

**Affiliations:** 1Changchun Institute of Optics, Fine Mechanics and Physics (CIOMP), Chinese Academy of Sciences, Changchun 130033, China; 2University of Chinese Academy of Sciences, Beijing 100039, China

**Keywords:** achromatic metalens, topology optimization, focal efficiency, polarization conversion efficiency

## Abstract

Metalens, composed of arrays of nano-posts, is an ultrathin planar optical element used for constructing compact optical systems which can achieve high-performance optical imaging by wavefront modulating. However, the existing achromatic metalenses for circular polarization possess the problem of low focal efficiency, which is caused by the low polarization conversion efficiencies of the nano-posts. This problem hinders the practical application of the metalens. Topology optimization is an optimization-based design method that can effectively extend the degree of design freedom, allowing the phases and polarization conversion efficiencies of the nano-posts to be taken into account simultaneously in the optimization procedures. Therefore, it is used to find geometrical configurations of the nano-posts with suitable phase dispersions and maximized polarization conversion efficiencies. An achromatic metalens has a diameter of 40 μm. The average focal efficiency of this metalens is 53% in the spectrum of 531 nm to 780 nm by simulation, which is higher than the previously reported achromatic metalenses with average efficiencies of 20~36%. The result shows that the introduced method can effectively improve the focal efficiency of the broadband achromatic metalens.

## 1. Introduction

As a planar optical element developed from metamaterials, the metalens can realize flexible modulation of light in the subwavelength scale, which has promising prospects in building lightweight and compacting optical systems when compared to conventional lenses [1]. It can be used in many applications [2] such as beam generators [3,4], optical holographic imaging [5,6], and virtual shaping [7,8]. However, the metalens has a dispersion problem which is inherent to the diffractive element [9].

To solve this problem, the parameter optimization method has been used to design achromatic metalens. This method is used to change the values of the geometric parameters of the nano-posts, in order to establish a library, and then to assemble the achromatic metalens based on the phase matching principle. An achromatic metalens [10] was designed based on the propagation phase modulation and work at three independent wavelengths of 1300 nm, 1550 nm, and 1800 nm, with the corresponding focal efficiencies of 15%, 10%, and 21%, respectively. Different nano-posts, including square nano-posts [11,12], circular nano-posts [13,14], hollow circular nano-posts [15], and nano-posts with a mixture of shapes [16], were likewise used to design polarization-independent metalenses in the continuous spectrum. The combination modulation [17,18,19,20,21,22,23,24] of the geometric phase and propagation phase was used to design achromatic metalenses by using rectangular nano-posts, with advantages in processing technology. However, the focal efficiencies of these metalenses did not exceed 40%. The low focal efficiencies are due to the orthogonal transformation of circular polarization states and the low polarization conversion efficiencies of the nano-posts. The polarization conversion efficiency is defined as the ratio of the field intensity of orthogonal light in the scattered field to the incident field. These are caused by the insufficiency of the design freedom of the nano-posts, such as the geometric parameters of the rectangular nano-posts. The insufficiency problem of the design freedom hinders the maintenance of maximum polarization conversion efficiency of the nano-post, while satisfying the required phase dispersion in the current design methods of achromatic metalenses.

Topology optimization originated from Michell’s [25] truss design research in 1904. It has been applied in electromagnetic waves, including beam splitters [26,27], photonic crystals [28,29], and metamaterials [30,31,32,33]. Topology optimization has been used in the global optimization of the metalens to improve its performance, for example the high-efficiency achromatic metalens [34], but is rarely used in the nanopillar optimization of the metalens. Different from the parameter optimization method, topology optimization is used to find the optimal distribution of materials in a given design domain according to the objective function and constraints. It is equivalent to extending the design freedom, which is beneficial to find the optimized nano-post with suitable phase dispersion and maximum polarization conversion efficiency. Therefore, we use the topology optimization to optimize the nano-posts and assemble the high-efficiency achromatic metalens based on the library composed of the obtained nano-posts.

## 2. Materials and Methods

An achromatic metalens can convert the incident plane wave of any wavelength in the working spectrum into the convergent spherical wave and make it focus on the same focal spot (Figure 1). Therefore, the phase profile on the metalens must satisfy the following condition [35]:(1)$$\varphi \left(r,\lambda \right)=-\frac{2\pi }{\lambda }\left(\sqrt{r^{2}+f^{2}}-f\right)+C\left(\lambda \right).$$
where *λ* is the wavelength of the incident wave, *r* is the radial coordinate on the metalens, and *f* is the focal length; *C*(*λ*) is an additional phase shift as a spectral degree of freedom, which is linear to the reciprocal of the wavelength *λ*. The phase profile can be satisfied by controlling the arrangement of TiO_2_ nano-posts (Figure 2a).

There are two approaches for the phase modulation of the achromatic metalens. The first is the propagation phase modulation. It is favorable to design high-efficiency metalens with the absence of orthogonal polarization conversion. However, the dual constraints of phase and phase dispersion severely elongate the nano-posts, which poses a problem for processing technology. The second approach is the combination modulation of the geometric phase and propagation phase, which reduces the processing challenges due to phase dispersion as the only physical quantity required to be considered in the optimization process of the nano-post. However, the achromatic metalens possess the problem of low focal efficiency. In consideration of processing technology, the second approach was adopted in our work so that the focal efficiency of orthogonal circularly polarized light could be enhanced to the greatest extent. Thus, in the working spectrum, Equation (1) can be split into two parts for the second approach [23]:(2)$$\varphi \left(r,\lambda \right)=\varphi \left(r,\lambda_{d}\right)+\Delta \varphi \left(r,\lambda \right).$$
where *λ_d_* is the design wavelength. The first term *φ*(*r*,*λ_d_*) on the right side of Equation (2) is the target phase profile at the design wavelength *λ_d_*, independent of the wavelength *λ*. It can be satisfied by rotating the nano-posts based on the geometric phase principle. For the left-handed circularly polarized incident light [1, *i*]^T^, the transmitted electric field can be described by the Jones vector:(3)$$E_{t}=\frac{t_{u}+t_{v}}{2}\left[\begin{array}{c} 1\\ i \end{array}\right]+\frac{t_{u}-t_{v}}{2}\exp\left(i\cdot 2\theta \right)\left[\begin{array}{c} 1\\ -i \end{array}\right].$$
where *t_u_* and *t_v_* represent complex transmission coefficients, with the incident light polarized along the long and short axes of the cross section of the nano-post, respectively; and *θ* is the rotation angle of the nano-post. In addition to the origin left-handed circularly polarized light with a complex amplitude of (*t_u_* + *t_v_*)/2, the scattered field also carries cross-polarized light with complex amplitude of (*t_u_* + *t_v_*)/2exp(*i*·2*θ*), where 2*θ* is a wavelength-independent geometric phase related to the rotation angle *θ* of the nano-post. The second term Δ*φ*(*r,λ*) in Equation (2) represents the phase difference between arbitrary wavelength and design wavelength:(4)$$\Delta \varphi \left(r,\lambda \right)=-2\pi \left(\frac{1}{\lambda }-\frac{1}{\lambda_{d}}\right)\left(\sqrt{r^{2}+f^{2}}-f\right).$$
(5)$$\frac{d\Delta \varphi }{d\left(2\pi c/\lambda \right)}=-\frac{1}{c}\left(\sqrt{r^{2}+f^{2}}-f\right).$$

The first equation in Equation (4) shows that Δ*φ*(*r,λ*) is a linear function of the reciprocal wavelength 1/*λ*. The second equation is the group delay, which is independent of the wavelength *λ* but dependent on the radial coordinate *r* on the metalens. Since the additional geometric phase is independent of the wavelength and does not interfere with the propagation phase, the group delay matching can only be achieved by adjusting the broadband propagation phase, which is related to the structure of the nano-post. Then, the optimization of the nano-post for the specific group delay is required.

## 3. Topology Optimization of Nano-Post

Topology optimization is used to mesh the design area, to determine whether the material in the grid cell should be retained or removed by numerical analysis. The material in every grid cell can be regarded as a degree of design freedom, increasing the design freedom of the nano-post. Thus, topology optimization can be used to optimize the structure of the nano-post, aiming for the most suitable phase and maximum polarization conversion efficiency. If group delay is the goal of the optimization model, the objective function can be written as φ(λ_1_) − φ(λ_2_). However, this objective function is ill-posed. An additional undetermined basic phase φ_0_ at design wavelength λ_d_ is then introduced to control φ(λ_1_) and φ(λ_2_) to solve this problem. Therefore, the deviations between the current phase and the target phase at the preset wavelengths are used to define the objective function instead of the group delay. In consideration of the problem that the phase reset characteristic will cause an inaccurate interpolation calculation, the sum of the least square deviations of complex amplitudes at multiple wavelengths is selected as the objective function:(6)$$\begin{array}{l} \min\;F={\sum_{m}\left|E_{srm}-\alpha_{m}e^{i\varphi_{tm}}\right|}^{2},\;\mathrm{in}\;\Omega_{d};\\ s.t.\;\nabla \times \left[\mu_{r}^{-1}\nabla \times \left(E_{sm}+E_{im}\right)\right]-k_{0}^{2}\varepsilon_{r}\left(E_{sm}+E_{im}\right)=0,\;\mathrm{in}\;\Omega \;;\\ \nabla \cdot E_{sm}=0,\;\mathrm{in}\;\Omega \;. \end{array}$$
where *k_0_ = 2π/λ* is the free space wave number; *ε_r_* and *μ_r_* are the relative permittivity and relative permeability of the nano-post, respectively; **E***_sm_* and **E***_im_* are the scattered and background field at *m*-th wavelength, respectively; *m* = 1, 2, 3 … represents the case for the *m*-th wavelength; **E***_srm_* is the electric field of the cross-polarized light; *α_m_* exp(*i*·*φ_tm_*) is the target electric field of the cross-polarized light; *α_m_* and *φ_tm_* represent the amplitude and target phase, respectively. The amplitude *α_m_* is introduced as a parameter to be scanned to reduce the objective function value. Specifically, it is scanned downward from 0.9 to find the maximum amplitude value that can make the objective function value approach zero. The target phase *φ_tm_* is calculated according to the target group delay and the basic phase *φ*_0_ at the design wavelength, which can take any value and be realized by the subsequent structural rotations. Thus, the basic phase *φ*_0_ needs to be scanned as an additional parameter to make the optimized group delay closer to the target group delay value.

The density method of topology optimization [36] is used to optimize the nano-post, where the density of dielectric material is treated as the design variable. To ensure robust evolution, the design variable is sequentially filtered and projected to derive the material density representing the geometrical configuration of the nano-posts. The filtered design variable is calculated using the PDE filter [37]. The filtered design variable can have an intermediate value, resulting in gray structure boundaries. To solve this problem, the filtered design variable is projected [38,39]. On the material interpolation, linear interpolation formula is used for air and dielectric materials TiO_2_:(7)$$\varepsilon_{r}\left(\gamma_{p}\right)=\varepsilon_{ra}+\gamma_{p}\left(\varepsilon_{rm}-\varepsilon_{ra}\right),\;\mathrm{in}\;\Omega_{d}.$$
where *ε_ra_* and *ε_rm_* are the relative permittivities of air and dielectric materials TiO_2_, respectively; *γ_p_* is the material density.

In this optimization model, the governing equation is firstly solved with the current design variable, and then the design variable is evolved according to the adjoint sensitivity, which is the first order variational of the objective function to the design variable:(8)$$\delta \hat{J}=\int_{\Omega_{d}}-\gamma_{fa}\delta \gamma d\Omega_{d}.$$
where *γ_fa_* is the adjoint variable of the filtered design variable. The topology optimization procedure is shown in the flowchart in Figure 3, where the finite element method is used to solve the wave equation and the design variable is evolved using the method of moving asymptotes. For more details of topology optimization refer to Appendix A.

## 4. Results and Discussion

Twenty values of the basic phase *φ*_0_ at the designed wavelength of 600 nm are uniformly sampled in the range of 0–2π and substituted into the topology optimization model of the nano-posts (Figure 2a). Taking the nano-post with the target group delay of 7 fs as an example, the simulated group delay and polarization conversion efficiency of the topologically optimized nano-post are obtained (Figure 4a). When the basic phase *φ_0_* is 0.3π, the group delay of the optimized nano-post 6.98 fs is closer to the target group delay, and the polarization conversion efficiency of the optimized nano-post is 88% at the designed wavelength. Figure 4b shows the phase of the optimized nano-post with the basic phase *φ*_0_ of 0.3π is linear to the frequency in the spectrum from 531 nm to 780 nm, and the broadband polarization conversion efficiency is no less than 40%. Therefore, this optimized nano-post can be extracted and placed at the position where the required group delay is 7 fs.

Different values of group delay are put into the topology optimization model of the nano-posts and the optimized nano-posts are entered into a library (some of them are shown in Figure 5). For more details of nano-posts in the library refer to Appendix A. Based on this established library, an achromatic metalens is designed with a diameter of 40 μm and NA of 0.1 according to the desired distribution of the group delay and phase at design wavelength, and the discrete phase realized by the nano-posts is consistent with the ideal phase profile (Figure 6). Figure 7a shows the focal spots on the designed focal length of 178.50 μm at several discrete wavelengths in the range of 531 nm to 780 nm. The maximum and minimum focal lengths of these wavelengths are 187.07 μm and 164.37 μm, respectively. The differences from the designed focal length are within 14.12 μm with the relative error of 7.91%. The focal efficiency is the ratio of the Poynting vector integral of the cross-polarized light within three times FWHM to the Poynting vector integral of the transmitted light. Through the above definition, the focal efficiencies of the achromatic metalens are calculated at nine wavelengths in the working spectrum (Figure 7d). The maximum and minimum focal efficiencies are 58% and 44%, respectively. In the whole working spectrum, the average efficiency is approximately 53%, higher than the previously reported achromatic metalens with average efficiencies of 20~36% [19,40]. When compared with other achromatic metalenses, the high focal efficiency of the achromatic metalens is mainly because of the high polarization conversion efficiency of the nano-posts. Further, another achromatic metalens is also designed with a diameter of 20 μm and NA of 0.22. The maximum relative focal length error of the metalens is 7.97%, and the average efficiency for the spectrum from 531 nm to 780 nm is 56%, which are all close values when compared with the performance of the achromatic metalens with NA of 0.11.

## 5. Conclusions

When targeting the problem of low focal efficiency of achromatic metalenses, we used the topology optimization method to optimize the structures of TiO_2_ nano-posts and derive the achromatic metalenses with an average focal efficiency of 53% in the spectrum from 531 nm to 780 nm. A library of the TiO_2_ nano-posts was derived by solving a topology optimization model to achieve maximum broadband focal efficiency at each group delay value. Based on the library, we assembled an achromatic metalens with a diameter of 40 μm and NA of 0.11 by combining the geometric phase and the propagation phase modulation. The maximum focal efficiency of the achromatic metalens was 58%. Topology optimization greatly increased the design freedom of the TiO_2_ nano-posts. It effectively solved the problem that the group delay and the polarization conversion efficiency cannot be taken into account simultaneously due to insufficient design freedom in the previous parameter optimization method. This work can be of significance in the promotion of high lightweight and high integration of broadband in the micro-imaging system.

## Figures and Tables

**Figure 1 nanomaterials-13-00890-f001:**
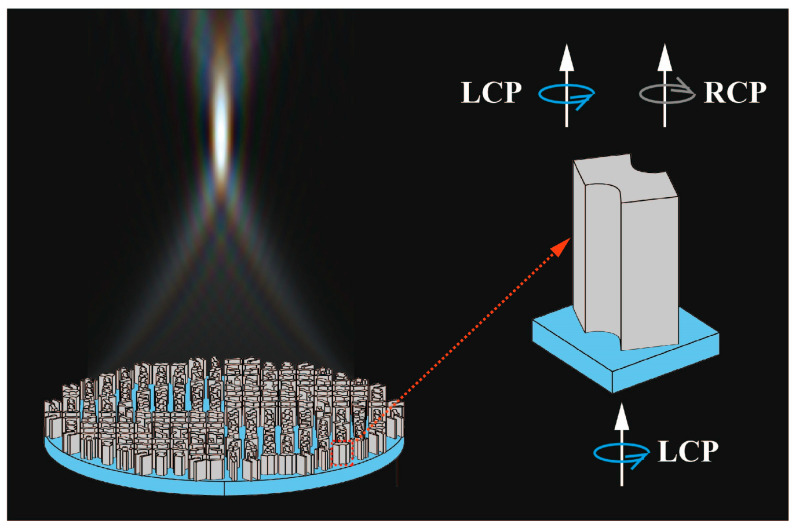
Sketch for the cross-section of an achromatic metalens.

**Figure 2 nanomaterials-13-00890-f002:**
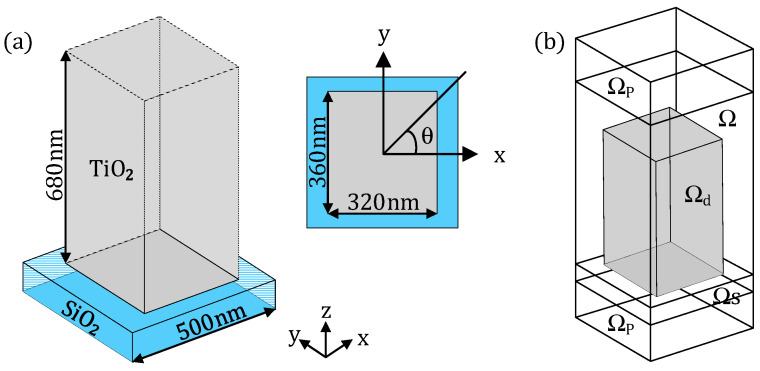
(**a**) The structure unit diagram of the achromatic metalens, where the gray rectangular block is the TiO_2_ nano-post; (**b**) sketch for the computational domain Ω. The upper and lower domains Ω_p_ are the perfectly matched layers (PMLs), which are used to suppress the boundary reflection. The substrate domain is Ω_s_, and the grey section is the design domain Ω_d_.

**Figure 3 nanomaterials-13-00890-f003:**
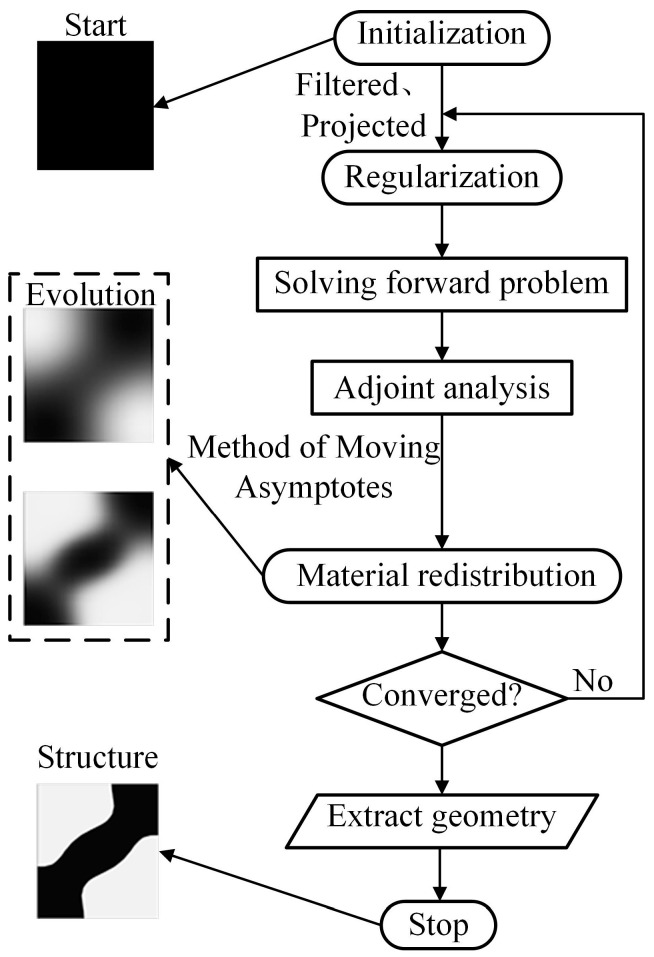
Flowchart for the topology optimization procedure.

**Figure 4 nanomaterials-13-00890-f004:**
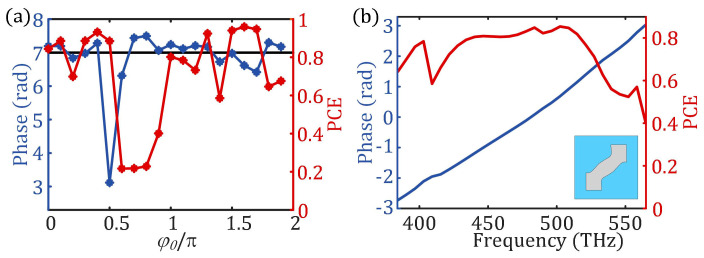
(**a**) The group delay (blue line) and polarization conversion efficiency (PCE) (red line) of the topology optimized nano-post under different basic phases *φ*_0_ at the design wavelength; (**b**) the phase (blue line) and polarization conversion efficiency (red line) of the structure unit (vertical view) composed of SiO_2_ substrate (blue area) and TiO_2_ nano-post (gray area) obtained by topology optimization with a basic phase of 0.3π in the spectrum from 531 nm to 780 nm.

**Figure 5 nanomaterials-13-00890-f005:**
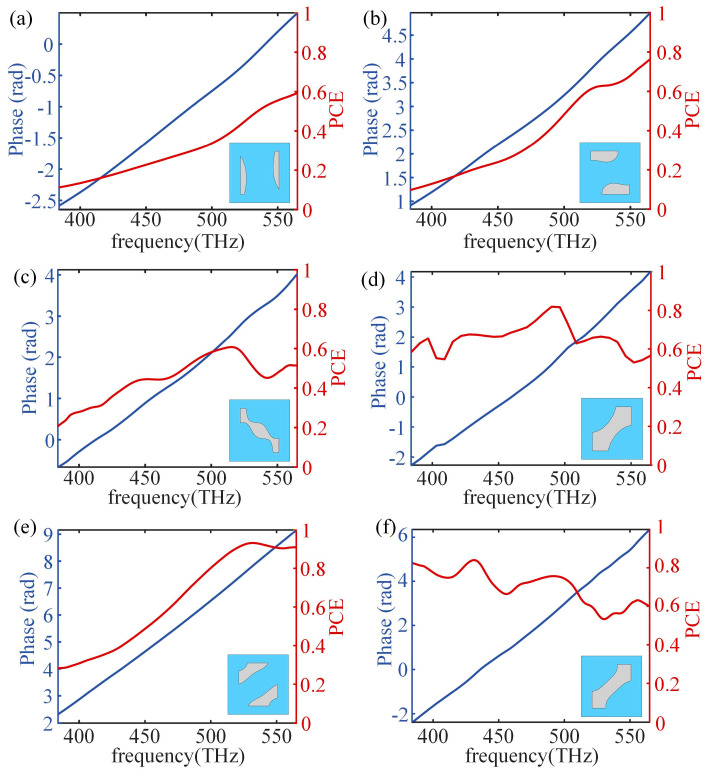
(**a**–**f**) The phase and polarization conversion efficiency (PCE) of some structure units (vertical view) composed of SiO_2_ substrates (blue area) and TiO_2_ nano-posts (gray area) in the library.

**Figure 6 nanomaterials-13-00890-f006:**
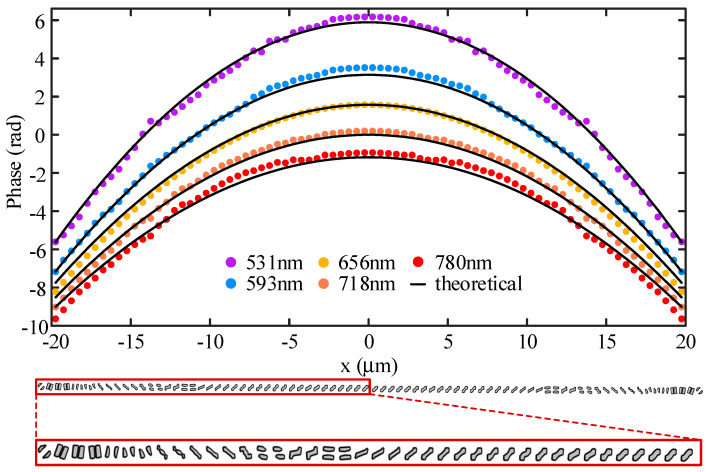
The actual discrete phase realized by the topologically optimized nano-posts (colored dot) and the ideal phase profile (black line) on the achromatic metalens, the bottom of which is the oblique view of the overall structure of the achromatic metalens along with the radius direction and top view of each local magnification.

**Figure 7 nanomaterials-13-00890-f007:**
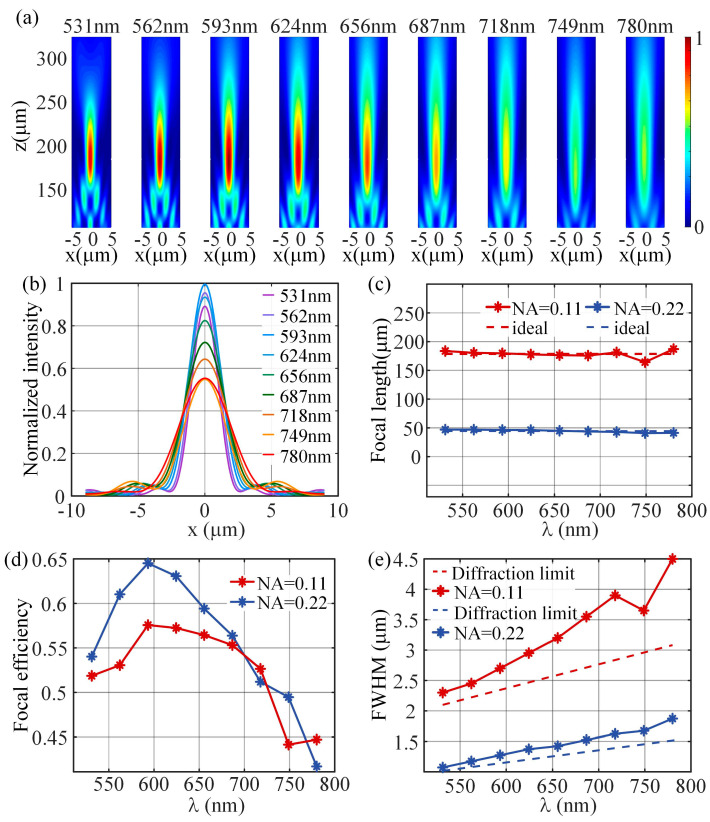
(**a**) Normalized electric field near the focal plane in the spectrum from 531 nm to 780 nm; (**b**) normalized point spread function (PSF); (**c**–**e**) the focal lengths (**c**), focal efficiencies (**d**), and FWHM (**e**) of the two metalenses with different NA values in the spectrum 531 nm~780 nm.

## Data Availability

Not applicable.

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
