# Peer review of "High-Efficiency Achromatic Metalens Topologically Optimized in the Visible"

_nanomaterials, 2023, doi:10.3390/nano13050890_

Round 1

Reviewer 1 Report

The reviewer suggests that the authors continue further advancement for making broad applivations.

Author Response

Dear editor and reviewers

Re: Manuscript ID: nanomaterials-2208070 and Title: High-efficiency achromatic metalens topologically optimized in visible

Thank you for your precious comments concerning our manuscript entitled “High-efficiency achromatic metalens topologically optimized in visible”(ID: nanomaterials-2208070). Those comments are all valuable and very helpful for revising and improving our paper, as well as the important guiding significance to our researches. We have studied comments carefully and have made correction which we hope meet with approval. Revised portion are marked in red in the paper. The main correction in the manuscript and the responds to the reviewer’s comments are as flowing.

Thank you for your careful review. We really appreciate your efforts in reviewing our manuscript during this unprecedented and challenging time. Your careful review has helped to make our study clearer and more comprehensive.

Sincerely,

Lijuan Zhang

The responds to the reviewer’s comments:

Point 1: The reviewer suggests that the authors continue further advancement for making broad applivations.

Response 1: We are grateful for the suggestion. The achromatic metalens in this manuscript is difficult to process due to our current processing level. However, we are now looking for a partner who can manufacture this metalens, or we can improve the topology optimization model so that we can manufacture achromatic metalens for beam generators, holographic imaging, virtual shaping, etc. 

Reviewer 2 Report

The manuscript describes a design and simulation study of a metasurface achromatic lens that is optimized using a specific topological method. I think the fact that it is a design and simulation study should be stated in the abstract and/or title of the manuscript.  TiO2 and SiO2 materials are assumed for the design, and the authors claim that their design is suitable for fabrication. I am interested in the topological design method as the authors claim it offers additional degrees of freedom in the design. I would like to see a more detailed introduction concerning this method. 

I find the introduction to be quite brief. The authors claim the focal efficiency is greater for their design than for other reported designs.  I am not sure why that is the only measure of effectiveness for the lenses reported. What is the overall power throughput predicted for the design?

Given that the main point of the manuscript is the design of a metasurface with specific properties, I expect that detailed design information should be available to readers. It is not evident that this is the case. 

Some of the English language needs editing for improved clarity.  The figures are fine. There is a weird symbol on line 219.

Author Response

Dear editor and reviewers

Re: Manuscript ID: nanomaterials-2208070 and Title: High-efficiency achromatic metalens topologically optimized in visible

Thank you for your precious comments concerning our manuscript entitled “High-efficiency achromatic metalens topologically optimized in visible”(ID: nanomaterials-2208070). Those comments are all valuable and very helpful for revising and improving our paper, as well as the important guiding significance to our researches. We have studied comments carefully and have made correction which we hope meet with approval. Revised portion are marked in red in the paper. The main correction in the paper and the responds to the reviewer’s comments are as flowing.

Thank you for your careful review. We really appreciate your efforts in reviewing our manuscript during this unprecedented and challenging time. Your careful review has helped to make our study clearer and more comprehensive.

Sincerely,

Lijuan Zhang

The responds to the reviewer’s comments:

Point 1: The manuscript describes a design and simulation study of a metasurface achromatic lens that is optimized using a specific topological method. I think the fact that it is a design and simulation study should be stated in the abstract and/or title of the manuscript.  TiO2 and SiO2 materials are assumed for the design, and the authors claim that their design is suitable for fabrication. I am interested in the topological design method as the authors claim it offers additional degrees of freedom in the design. I would like to see a more detailed introduction concerning this method. 

Response 1: We are grateful for the comment. We have changed [An achromatic metalens is derived with a diameter of 20mm and the average focal efficiency of 53% in the spectrum from 531nm to 780nm] to [An achromatic metalens is derived with a diameter of 40mm. The average focal efficiency of this metalens is 53% in the spectrum from 531nm to 780nm by simulation] on line 18-19. Topology optimization is to mesh the design area, determine whether materials in the grid cell should be retained or removed by numerical analysis. The materials in every grid cell can be regarded as a degree of design freedom. It increases the design freedom of the nano-post. And we have added a more detailed interpretation regarding topology optimization in supporting information.

Point 2: I find the introduction to be quite brief. The authors claim the focal efficiency is greater for their design than for other reported designs.  I am not sure why that is the only measure of effectiveness for the lenses reported. What is the overall power throughput predicted for the design?

Response 2: Thank you for your comment. The focal efficiency is not the only measure of the reported metalens. The measures of the metalens are resolution ratio, chromatic difference and focal efficiency, etc. We only introduce focal efficiencies of the reported achromatic metalenses, because our research is mainly focused on the focal efficiency of achromatic metalens. The average total focusing efficiency of the achromatic metalens with a diameter of 40mm is 52.58%. The average total focusing efficiency is defined as the average of total focal efficiencies at 9 simulated wavelengths, where total focal efficiency is the ratio of the Poynting vector integral of the cross-polarized light within 3 times FWHM to the Poynting vector integral of the incident light.

Point 3: Given that the main point of the manuscript is the design of a metasurface with specific properties, I expect that detailed design information should be available to readers. It is not evident that this is the case. 

Response 3: We deeply appreciate your comment. According to the comment of the reviewer, we add a more detailed design information in supporting information. We have added the information “according to the desired distribution of the group delay and phase” (Lines 192-193).

Point 4: Some of the English language needs editing for improved clarity.  The figures are fine. There is a weird symbol on line 219.

Response 4: We apologize for the language problems in the original manuscript. For this language problem, “40 m” is revised as “40mm” and it is highlighted on line 234.

Reviewer 3 Report

Manuscript ID: nanomaterials- 2208070

Title: High-efficiency achromatic metalens topologically optimized 2 in visible

Reviewer’s comments:

The paper presents studies regarding a method that can improve the focal efficiency of broadband (531-780 nm domain) achromatic metalens. I have the following observations:

1. The subject studied by the authors is in rapid evolution. Thus, a reference to recent paper treating the dielectric metalens could improve the presentation:

a)Pan, M., Fu, Y., Zheng, M., Chen, H., Zang, Y., Duan, H., Li, Q., Qiu, M. and Hu, Y., 2022. Dielectric metalens for miniaturized imaging systems: progress and challenges. Light: Science & Applications, 11(1), p.195.

Also, the authors should refer to an interesting point of view regarding the metasurface that exibit bifunctional behavior, that can be found in the recent paper:

b) Danila, O. and Manaila-Maximean, D., 2021. Bifunctional metamaterials using spatial phase gradient architectures: Generalized reflection and refraction considerations. Materials, 14(9), p.2201.

2. The authors should define the physical quantities that are presented in the graphical representations (polarization conversion efficiency, focal efficiency, average efficiency).  

3. Some small typo mistakes can still be found in the manuscript, such as:

r. 73  the “first” one

r.78   “only to”

and others.

My recommendation is: Minor revision of the manuscript.

Author Response

Dear editor and reviewers

Re: Manuscript ID: nanomaterials-2208070 and Title: High-efficiency achromatic metalens topologically optimized in visible

Thank you for your precious comments concerning our manuscript entitled “High-efficiency achromatic metalens topologically optimized in visible”(ID: nanomaterials-2208070). Those comments are all valuable and very helpful for revising and improving our paper, as well as the important guiding significance to our researches. We have studied comments carefully and have made correction which we hope meet with approval. Revised portion are marked in red in the paper. The main correction in the paper and the responds to the reviewer’s comments are as flowing.

Thank you for your careful review. We really appreciate your efforts in reviewing our manuscript during this unprecedented and challenging time. Your careful review has helped to make our study clearer and more comprehensive.

Sincerely,

Lijuan Zhang

The responds to the reviewer’s comments:

Point 6: The subject studied by the authors is in rapid evolution. Thus, a reference to recent paper treating the dielectric metalens could improve the presentation:

  1. Pan, M., Fu, Y., Zheng, M., Chen, H., Zang, Y., Duan, H., Li, Q., Qiu, M. and Hu, Y., 2022. Dielectric metalens for miniaturized imaging systems: progress and challenges. Light: Science & Applications11(1), p.195.

Response 6: Thank you for your comments. This paper has been very helpful to our manuscript, and we also introduce this paper on line 32 and highlight this revision.

Point 7: Also, the authors should refer to an interesting point of view regarding the metasurface that exibit bifunctional behavior, that can be found in the recent paper:

  1. b) Danila, O. and Manaila-Maximean, D., 2021. Bifunctional metamaterials using spatial phase gradient architectures: Generalized reflection and refraction considerations. Materials14(9), p.2201.

Response 7: We are grateful for the comments. We read this paper and find that this paper has inspired our follow-up work. So we introduce this paper on line 30, and highlight this revision.

Point 8: The authors should define the physical quantities that are presented in the graphical representations (polarization conversion efficiency, focal efficiency, average efficiency). 

Response 8: We deeply appreciate your comments. According to the comment, we have added a more interpretation regarding these physical quantities. The polarization conversion efficiency is the ratio of the field intensity of orthogonal light in the scattered field to the incident field. It is added on line 46-47. The focal efficiency is the ratio of the Poynting vector integral of the cross-polarized light within 3 times FWHM to the Poynting vector integral of the transmitted light. The average efficiency is the average of focal efficiencies at 9 simulated wavelengths. We change the efficiency in the figure to polarization conversion efficiency or focal efficiency on line112-113, line 181-182, line 211-212, line 222-223.

Point 9: Some small typo mistakes can still be found in the manuscript, such as:

  1. 73  the “first” one

r.78   “only to”

and others.

Response 9: We apologize for the poor language of our manuscript. We worked on the manuscript for a long time and the repeated addition and removal of sentences and sections obviously led to poor readability. We have now worked on both language and readability. We really hope that the flow and language level have been substaintially improved. We change [The one is the progation phase modulation] to [The first one is the progation phase modulation] on line 77 and highlight it. And we change [due to only phase dispersion required to be considered] to [due to phase dispersion as the only physical quantity required to be considered] on line81-82 and highlight it.
